# M3CoL: Harnessing Shared Relations via Multimodal Mixup Contrastive Learning for Multimodal Classification

**Raja Kumar**[‡1], **Raghav Singhal**[‡1], **Pranamya Kulkarni**[1], **Deval Mehta**[2], and **Kshitij Jadhav**[1]

[1]Indian Institute of Technology Bombay, Mumbai, India
[2]AIM for Health Lab, Department of Data Science & AI, Monash University, Australia

## Abstract

Deep multimodal learning has shown remarkable success by leveraging contrastive learning to capture explicit one-to-one relations across modalities. However, real-world data often exhibits shared relations beyond simple pairwise associations. We propose **M3CoL**, a **M**ulti**m**odal **M**ixup **Co**ntrastive **L**earning approach to capture nuanced *shared relations* inherent in multimodal data. Our key contribution is a Mixup-based contrastive loss that learns robust representations by aligning mixed samples from one modality with the corresponding samples from other modalities. For multimodal classification tasks, we introduce a framework that integrates a fusion module with unimodal prediction modules for auxiliary supervision during training, complemented by our proposed Mixup-based contrastive loss. Through extensive experiments on diverse datasets (N24News, ROSMAP, BRCA, and Food-101), we demonstrate that **M3CoL** effectively captures shared multimodal relations and generalizes across domains. It outperforms state-of-the-art methods on N24News, ROSMAP, and BRCA, while achieving comparable performance on Food-101. Our work highlights the significance of learning shared relations for robust multimodal learning, opening up promising avenues for future research.

## 1 Introduction

In the era of abundant multimodal data, it is crucial to equip artificial intelligence with multimodal capabilities [1]. At the heart of advancements in multimodal learning is contrastive learning, which maximizes similarity for positive pairs and minimizes it for negative pairs, making it practical for multimodal representation learning. CLIP [2] is a prominent example that employs contrastive learning to understand the direct link between paired modalities and seamlessly maps images and text into a shared space for cross-modal understanding. However, traditional contrastive learning methods often overlook shared relationships between samples across different modalities, which can result in the learning of representations that are not fully optimized for capturing the underlying connections between diverse data types. These methods focus on distinguishing between positive and negative pairs of samples, typically treating each instance as an independent entity. They tend to disregard the rich, shared relational information that could exist between samples within and across modalities.

While traditional contrastive learning methods treat paired modalities as positive samples and non-corresponding ones as negative, they often overlook shared relations between different samples. As shown in the left panel of Figure 1 (Left panel), classical contrastive learning approach assumes perfect one-to-one relations between modalities, which is rare in real-world data. For example, shared

---

[‡]Equal Contributions. Author ordering determined by coin flip over Google Meet.
Our code is available at: https://github.com/RaghavSinghal10/M3CoL.

38th Conference on Neural Information Processing Systems (NeurIPS 2024).

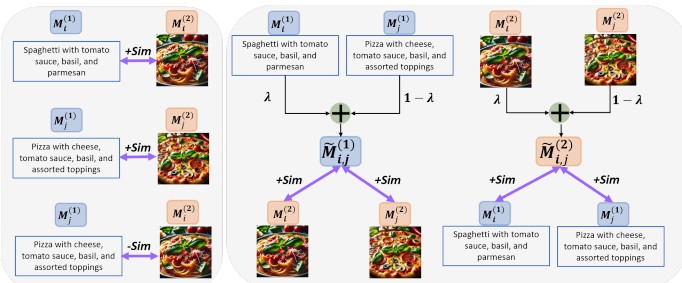

Figure 1: Comparison of traditional contrastive and our proposed M3Co loss. $\mathbf{M}_i^{(1)}$ and $\mathbf{M}_i^{(2)}$ denote representations of the $i$-th sample from modalities 1 and 2, respectively. Traditional contrastive loss (left panel) aligns corresponding sample representations across modalities. M3Co (right panel) mixes the $i$-th and $j$-th samples from modality 1 and enforces the representations of this mixture to align with the representations of the corresponding $i$-th and $j$-th samples from modality 2, and vice versa.

elements in images or text can relate even across separate samples, as illustrated by the elements like *"tomato sauce"* and *"basil"* in Figure 1. Our approach, illustrated in the right panel of Figure 1, goes beyond simple pairwise alignment by capturing shared relationships across mixed samples. By creating newer data points through convex combinations of data points our method effectively models complex relationships, such as imperfect bijections [3], enhancing multimodal performance.

Our approach builds upon the success of data augmentation techniques such as Mixup [4] and their variants [5–7], which have proven beneficial for enhancing learned feature spaces, improving both robustness and performance. Mixup trains models on synthetic data created through convex combinations of two datapoint-label pairs [8]. These techniques are particularly valuable in low sample settings, as they help prevent overfitting and the learning of ineffective shortcuts [9, 10], common in contrastive learning. Building on the success of recent Mixup strategies [11–13] and MixCo [14], we introduce M3Co, a novel approach that adapts and enhances contrastive learning principles to complex multimodal settings. M3Co modifies the CLIP loss to handle multimodal scenarios, addressing the problem of instance discrimination, where models overly focus on distinguishing individual instances instead of capturing relationships between modalities. M3Co eliminates instance discrimination and enhances robust representation learning by capturing shared relations. Our results demonstrate improvements in performance and generalization across a range of multimodal tasks.

## 2 Methodology

**Pipeline Overview.** Figure 2 depicts our framework, which comprises of three components: unimodal prediction modules, a fusion module, and a Mixup-based contrastive loss. We obtain latent representations (using learnable modality specific encoders $f^{(1)}$ and $f^{(2)}$) of individual modalities and fuse them (denoted by concatenation symbol '+') to generate a joint multimodal representation, which is optimized using a supervised objective (through classifier 3). The unimodal prediction modules provide additional supervision during training (via classifier 1 and 2). These strategies enable deeper integration of modalities and allow the models to compensate for the weaknesses of one modality with the strengths of another. The Mixup-based contrastive loss (denoted by $\mathcal{L}_{\text{M3Co}}$) updates the representations by capturing shared relations inherent in the multimodal data.

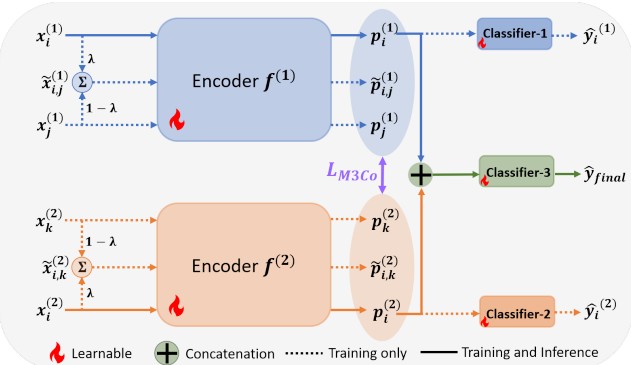

Figure 2: Architecture of our proposed M3CoL model.

**Multimodal Mixup Contrastive Learning.** Given a batch of $N$ multimodal samples, let $\mathbf{x}_i^{(1)}$ and $\mathbf{x}_i^{(2)}$ denote the $i$-th samples for the first and second modalities, respectively. The modality encoders, $f^{(1)}$ and $f^{(2)}$, generate the corresponding embeddings $\mathbf{p}_i^{(1)}$ and $\mathbf{p}_i^{(2)}$:

$$\mathbf{p}_i^{(1)} = f^{(1)}(\mathbf{x}_i^{(1)}), \quad \mathbf{p}_i^{(2)} = f^{(2)}(\mathbf{x}_i^{(2)}) \tag{1}$$

We generate a mixture, $\tilde{\mathbf{x}}_{i,j}^{(1)}$, of the samples $\mathbf{x}_i^{(1)}$ and $\mathbf{x}_j^{(1)}$ by taking their convex combination. Similarly, we generate a mixture, $\tilde{\mathbf{x}}_{i,k}^{(2)}$, using the convex combination of the samples $\mathbf{x}_i^{(2)}$ and $\mathbf{x}_k^{(2)}$ (Eq. 2). For the text modality, instead of directly mixing the raw inputs, we mix the text embeddings [15]. The mixing indices $j, k$ are drawn arbitrarily, without replacement, from $[1, N]$, for both the modalities. We mix both the modalities using a factor $\lambda \sim \text{Beta}(\alpha, \alpha)$. Based on the findings of [4], which demonstrated enhanced performance for $\alpha$ values between 0.1 and 0.4, we chose $\alpha = 0.15$ after experimenting with several values in this range. The mixtures are fed through the respective encoders to obtain the embeddings: $\tilde{\mathbf{p}}_{i,j}^{(1)}$, and $\tilde{\mathbf{p}}_{i,k}^{(2)}$ (Eq. 3).

$$\tilde{\mathbf{x}}_{i,j}^{(1)} = \lambda_i \cdot \mathbf{x}_i^{(1)} + (1 - \lambda_i) \cdot \mathbf{x}_j^{(1)}, \quad \tilde{\mathbf{x}}_{i,k}^{(2)} = \lambda_i \cdot \mathbf{x}_i^{(2)} + (1 - \lambda_i) \cdot \mathbf{x}_k^{(2)} \tag{2}$$

$$\tilde{\mathbf{p}}_i^{(1)} = \tilde{\mathbf{p}}_{i,j}^{(1)} = f^{(1)}(\tilde{\mathbf{x}}_{i,j}^{(1)}), \quad \tilde{\mathbf{p}}_i^{(2)} = \tilde{\mathbf{p}}_{i,k}^{(2)} = f^{(2)}(\tilde{\mathbf{x}}_{i,k}^{(2)}) \tag{3}$$

We generate embeddings for the entire batch $\tilde{\mathbf{p}}^{(1)}$ and $\tilde{\mathbf{p}}^{(2)}$, where the $i$-th elements, $\tilde{\mathbf{p}}_i^{(1)}$ and $\tilde{\mathbf{p}}_i^{(2)}$, correspond to $\tilde{\mathbf{p}}_{i,m_i}^{(1)}$ and $\tilde{\mathbf{p}}_{i,m_i}^{(2)}$, respectively. The unidirectional contrastive loss [9, 16–19] over $\mathbf{p}^{(2)}$ is conventionally defined as:

$$\mathcal{L}_{\text{sim-conv}}(\mathbf{p}^{(1)}, \mathbf{p}^{(2)}) = -\frac{1}{N} \sum_{i=1}^{N} \log \frac{\exp\left(\mathbf{p}_i^{(1)} \cdot \mathbf{p}_i^{(2)}/\tau\right)}{\sum\limits_{j=1}^{N} \exp\left(\mathbf{p}_i^{(1)} \cdot \mathbf{p}_j^{(2)}/\tau\right)} \tag{4}$$

where $\cdot$ indicates dot product and $\tau$ is a temperature hyperparameter. While this formulation is suitable for computing similarity among aligned samples from different modalities, our method requires flexibility to handle both aligned and non-aligned samples. To achieve this, we define the unidirectional multimodal contrastive loss between $\mathbf{p}_i^{(1)}$ and $\mathbf{p}_m^{(2)}$ over $\mathbf{p}^{(2)}$ as:

$$\mathcal{L}_{\text{sim}}(\mathbf{p}_i^{(1)}, \mathbf{p}^{(2)}; m) = -\log \frac{\exp\left(\mathbf{p}_i^{(1)} \cdot \mathbf{p}_m^{(2)}/\tau\right)}{\sum\limits_{j=1}^{N} \exp\left(\mathbf{p}_i^{(1)} \cdot \mathbf{p}_j^{(2)}/\tau\right)} \tag{5}$$

where $\mathbf{p}^{(1)}$ and $\mathbf{p}^{(2)}$ are $\mathcal{L}^2$ normalized, $\tau$ is a temperature hyperparameter, and $m$ is a sample index in $[1, N]$. Although the multimodal contrastive loss (Eq. 5) can learn indirect relations, it is insufficient for learning shared semi-positive relations between modalities. Therefore, we introduce a Mixup-based contrastive loss to capture these relations that promotes generalized learning, as this process is more nuanced than simply discriminating positives from negatives. Now, following standard works [2, 16–18], we make our loss bidirectional. We define this bidirectional Mixup contrastive loss M3Co for each modality (Eq. 6, 7) and the total M3Co loss as:

$$\mathcal{L}_{\text{M3Co}}^{(1)} = \frac{1}{N} \sum_{i=1}^{N} \left[ \lambda_i \cdot \mathcal{L}_{\text{sim}}(\tilde{\mathbf{p}}_{i,j}^{(1)}, \mathbf{p}^{(2)}; i) + (1 - \lambda_i) \cdot \mathcal{L}_{\text{sim}}(\tilde{\mathbf{p}}_{i,j}^{(1)}, \mathbf{p}^{(2)}; j) \right]$$

$$+ \frac{1}{N} \sum_{i=1}^{N} \left\{ \lambda_i \cdot \mathcal{L}_{\text{sim}}(\mathbf{p}_i^{(2)}, \tilde{\mathbf{p}}^{(1)}; i) + (1 - \lambda_i) \cdot \mathcal{L}_{\text{sim}}(\mathbf{p}_j^{(2)}, \tilde{\mathbf{p}}^{(1)}; i) \right\} \tag{6}$$

$$\mathcal{L}_{\text{M3Co}}^{(2)} = \frac{1}{N} \sum_{i=1}^{N} \left[ \lambda_i \cdot \mathcal{L}_{\text{sim}}(\tilde{\mathbf{p}}_{i,k}^{(2)}, \mathbf{p}^{(1)}; i) + (1 - \lambda_i) \cdot \mathcal{L}_{\text{sim}}(\tilde{\mathbf{p}}_{i,k}^{(2)}, \mathbf{p}^{(1)}; k) \right]$$

$$+ \frac{1}{N} \sum_{i=1}^{N} \left\{ \lambda_i \cdot \mathcal{L}_{\text{sim}}(\mathbf{p}_i^{(1)}, \tilde{\mathbf{p}}^{(2)}; i) + (1 - \lambda_i) \cdot \mathcal{L}_{\text{sim}}(\mathbf{p}_k^{(1)}, \tilde{\mathbf{p}}^{(2)}; i) \right\} \tag{7}$$

$\mathcal{L}_{\text{M3Co}}^{(1,2)} = \frac{1}{2}\left(\mathcal{L}_{\text{M3Co}}^{(1)} + \mathcal{L}_{\text{M3Co}}^{(2)}\right)$, where $\mathbf{p}^{(1)}$, $\tilde{\mathbf{p}}^{(1)}$, $\mathbf{p}^{(2)}$, and $\tilde{\mathbf{p}}^{(2)}$ are $\mathcal{L}^2$ normalized. Note that the parts of the loss functions in Eq. (6, 7) inside curly parantheses make them bidirectional. Mixup-based methods enhance generalization by capturing clean patterns in the early training stages but can eventually overfit to noise if continued for too long [20–22]. To address this, we implement a schedule that transitions from the Mixup-based M3Co loss to a non-Mixup multimodal contrastive loss. We design this transition so that the non-Mixup loss retains the ability to learn shared or indirect relationships between modalities. By using a bidirectional SoftClip-based loss [9, 16, 23], we relax the rigid one-to-one correspondence, allowing the model to capture many-to-many relations [23, 24]. The bidirectional **MultiS**oft**Clip** loss for each modality (Eq. 8, 9) and its combination is:

$$\mathcal{L}_{\text{MultiSClip}}^{(1)} = \frac{1}{N}\sum_{i=1}^{N}\sum_{l=1}^{N}\left[\frac{\exp\left(\mathbf{p}_i^{(1)} \cdot \mathbf{p}_l^{(1)}/\tau\right)}{\sum\limits_{t=1}^{N}\exp\left(\mathbf{p}_i^{(1)} \cdot \mathbf{p}_t^{(1)}/\tau\right)} \cdot \left(\mathcal{L}_{\text{sim}}(\mathbf{p}_i^{(2)}, \mathbf{p}^{(1)}; l) + \mathcal{L}_{\text{sim}}(\mathbf{p}_l^{(1)}, \mathbf{p}^{(2)}; i)\right)\right]$$

(8)

$$\mathcal{L}_{\text{MultiSClip}}^{(2)} = \frac{1}{N}\sum_{i=1}^{N}\sum_{l=1}^{N}\left[\frac{\exp\left(\mathbf{p}_i^{(2)} \cdot \mathbf{p}_l^{(2)}/\tau\right)}{\sum\limits_{t=1}^{N}\exp\left(\mathbf{p}_i^{(2)} \cdot \mathbf{p}_t^{(2)}/\tau\right)} \cdot \left(\mathcal{L}_{\text{sim}}(\mathbf{p}_i^{(1)}, \mathbf{p}^{(2)}; l) + \mathcal{L}_{\text{sim}}(\mathbf{p}_l^{(2)}, \mathbf{p}^{(1)}; i)\right)\right]$$

(9)

$\mathcal{L}_{\text{MultiSClip}}^{(1,2)} = \frac{1}{2}\left(\mathcal{L}_{\text{MultiSClip}}^{(1)} + \mathcal{L}_{\text{MultiSClip}}^{(2)}\right)$, where $\mathbf{p}^{(1)}$ and $\mathbf{p}^{(2)}$ are $\mathcal{L}^2$ normalized. The M3Co and MultiSClip losses for $M$ modalities is:

$$\mathcal{L}_{\text{M3Co}} = \sum_{i=1}^{M}\sum_{j>i}^{M}\mathcal{L}_{\text{M3Co}}^{(i,j)}, \mathcal{L}_{\text{MultiSClip}} = \sum_{i=1}^{M}\sum_{j>i}^{M}\mathcal{L}_{\text{MultiSClip}}^{(i,j)} \tag{10}$$

**Unimodal Predictions and Fusion.** The encoders produce latent representations for each of the $M$ modalities, serving as inputs to individual classifiers that generate modality-specific predictions. These representations are used for modality-specific supervision only during training. The unimodal prediction task, $\mathcal{L}_{\text{CE-Uni}}$, involves minimizing the cross-entropy loss $\mathcal{L}_{\text{CE}}$ between these predictions and the corresponding ground truth labels, for each modality. We merge the unimodal latent representations by concatenating them and pass the combined representation to the output classifier. These predictions serve as the final outputs used during inference. The multimodal prediction process, $\mathcal{L}_{\text{CE-Multi}}$, minimizes the cross-entropy loss between the predictions and the corresponding labels.

**Combined Learning Objective.** Our overall loss objective utilizes a schedule to combine our M3Co and MultiSClip loss functions weighted by a hyperparameter $\beta$, along with the unimodal and multimodal cross-entropy losses. We use M3Co for the first one-third [20] part of training, and then transition to MultiSClip. The end-to-end loss is defined as:

$$\mathcal{L}_{\text{Total}} = \beta \cdot \mathcal{L}_{\text{M3Co | MultiSClip}} + \mathcal{L}_{\text{CE-Uni}} + \mathcal{L}_{\text{CE-Multi}} \tag{11}$$

## 3 Experiments and Results

**Datasets and Implementation Details.** We evaluate on four diverse multimodal classification datasets: N24News [25], Food-101 [26], ROSMAP [27], and BRCA [27]. N24News and Food-101 are image-text classification datasets. ROSMAP and BRCA are medical datasets, each containing three modalities: DNA methylation, miRNA expression, and mRNA expression. We use a ViT [28] as the image encoder for N24News and Food-101. For N24News, the text encoder is a pretrained BERT/RoBERTa [29, 30], while we use a pretrained BERT as the text encoder for Food-101. The classifiers for the above two datasets are three layer MLPs with ReLU activations. For ROSMAP and BRCA, which are small datasets, we use two layer MLPs as feature encoders for each modality, and two layer MLPs as classifiers. Details and related work are presented in Appendix A.1 and A.5.

**Results.** The experimental results from Table 1, 2, 5, reveal the following findings: **(i)** M3CoL consistently outperforms all SOTA methods across all text sources on N24News when using the same encoders, beats SOTA on all evaluation metrics on ROSMAP and BRCA, and also achieves competitive results on Food-101; **(ii)** contrastive-based methods with any form of alignment demonstrate superior performance compared to other multimodal methods; **(iii)** our proposed M3CoL method,

which employs a contrastive-based approach with shared alignment, improves over the traditional contrastive-based models and the SOTA multimodal methods. We present a detailed analysis of the various components of our method in Table 6, and text-guided visualization in Appendix A.4.

| Method | Fusion | | Backbone | | ACC ↑ | | |
|---|---|---|---|---|---|---|---|
| | AGG | ALI | Image | Text | Headline | Caption | Abstract |
| Image-only | - | - | ViT | - | 54.1 (*no text source used*) | | |
| Text-only | - | - | - | BERT | 72.1 | 72.7 | 78.3 |
| UniConcat | Early | ✗ | ViT | BERT | 78.6 | 76.8 | 80.8 |
| UniS-MMC | Early | ✓ | ViT | BERT | 80.3 | 77.5 | 83.2 |
| M3CoL (Ours) | Early | ✓ | ViT | BERT | $80.8_{\pm 0.05}$ | $78.0_{\pm 0.03}$ | $83.8_{\pm 0.06}$ |
| Text-only | - | - | - | RoBERTa | 71.8 | 72.9 | 79.7 |
| UniConcat | Early | ✗ | ViT | RoBERTa | 78.9 | 77.9 | 83.5 |
| N24News | Early | ✗ | ViT | RoBERTa | 79.41 | 77.45 | 83.33 |
| UniS-MMC | Early | ✓ | ViT | RoBERTa | 80.3 | 78.1 | 84.2 |
| M3CoL (Ours) | Early | ✓ | ViT | RoBERTa | $80.9_{\pm 0.19}$ | $79.2_{\pm 0.08}$ | $84.7_{\pm 0.03}$ |

Table 1: Classification Accuracy (ACC) on N24News on three different text sources. AGG denotes early/late modality fusion, ALI indicates presence/absence of alignment. Our method consistently outperforms the state-of-the-art across all text sources and backbone combinations.

| Method | Fusion | | ROSMAP | | | BRCA | | |
|---|---|---|---|---|---|---|---|---|
| | AGG | ALI | ACC ↑ | F1 ↑ | AUC ↑ | ACC ↑ | WF1 ↑ | MF1 ↑ |
| GRidge | Early | ✗ | 76.0 | 76.9 | 84.1 | 74.5 | 72.6 | 65.6 |
| BPLSDA | Early | ✗ | 74.2 | 75.5 | 83.0 | 64.2 | 53.4 | 36.9 |
| BSPLSDA | Early | ✗ | 75.3 | 76.4 | 83.8 | 63.9 | 52.2 | 35.1 |
| MOGONET | Late | ✗ | 81.5 | 82.1 | 87.4 | 82.9 | 82.5 | 77.4 |
| TMC | Late | ✗ | 82.5 | 82.3 | 88.5 | 84.2 | 84.4 | 80.6 |
| CF | Early | ✗ | 78.4 | 78.8 | 88.0 | 81.5 | 81.5 | 77.1 |
| GMU | Early | ✗ | 77.6 | 78.4 | 86.9 | 80.0 | 79.8 | 74.6 |
| MOSEGCN | Early | ✗ | 83.0 | 82.7 | 83.2 | 86.7 | 86.8 | 81.1 |
| DYNAMICS | Early | ✗ | 85.7 | 86.3 | 91.1 | 87.7 | 88.0 | 84.5 |
| M3CoL (Ours) | Early | ✓ | $88.7_{\pm 0.94}$ | $88.5_{\pm 0.94}$ | $92.6_{\pm 0.59}$ | $88.4_{\pm 0.57}$ | $89.0_{\pm 0.42}$ | $86.2_{\pm 0.54}$ |

Table 2: Comparison of Classification Accuracy (ACC), Area Under the Curve (AUC), F1 score (F1) on ROSMAP, and Classification Accuracy (ACC), Weighted F1 score (WF1), and Micro F1 score (MF1) on BRCA datasets. AGG denotes early/late modality fusion, ALI indicates presence/absence of alignment. Our method significantly outperforms the state-of-the-art across all metrics.

**Discussion and Conlusions.** Aligning representations across modalities presents significant challenges due to the complex, often non-bijective relationships in real-world multimodal data [3]. These relationships can involve many-to-many mappings or even lack clear associations, as exemplified by linguistic ambiguities and synonymy in vision-language tasks. We propose M3Co, a novel contrastive-based alignment method that captures shared relations beyond explicit pairwise associations by aligning mixed samples from one modality with corresponding samples from others. Our approach incorporates Mixup-based contrastive learning, introducing controlled noise that mirrors the inherent variability in multimodal data, thus enhancing robustness and generalizability. The M3Co loss, combined with an architecture leveraging unimodal and fusion modules, enables continuous updating of representations necessary for accurate predictions and deeper integration of modalities. This method generalizes across diverse domains, including image-text, high-dimensional multi-omics, and data with more than two modalities. Experiments on four public multimodal classification datasets demonstrate the effectiveness of our approach in learning robust representations that surpass traditional multimodal alignment techniques.

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

# A Appendix

## A.1 Experimental and Dataset Details

**Experimental Details.** The results are reported as the average and standard deviation over three runs on Food-101 and N24News, and five runs on ROSMAP and BRCA. We use a grid search on the validation set to search for optimal hyperparameters. The temperature parameter for the M3Co and MultiSClip losses is set to 0.1. The corresponding loss coefficient $\beta$ is 0.1 to keep the loss value in the same range as the other losses. We use the Adam optimizer [31] for all datasets. For Food-101 and N24News, the learning rate scheduler is ReduceLROnPlateau with validation accuracy as the monitored metric, lr factor of 0.2, and lr patience of 2. For ROSMAP and BRCA, we use the StepLR scheduler with a step size of 250. For Food-101 and N24News, the maximum token length of the text input for the BERT/RoBERTa encoders is 512. Other hyperparameter details are provided in Table 3.

| Hyperparameter | N24News | Food-101 | ROSMAP | BRCA |
|---|---|---|---|---|
| Embedding dimension | 768 | 768 | 1000 | 768 |
| Classifier dimension | 256 | 256 | 1000 | 768 |
| Learning rate | $10^{-4}$ | $10^{-4}$ | $5 \cdot 10^{-3}$ | $5 \cdot 10^{-3}$ |
| Weight decay | $10^{-4}$ | $10^{-4}$ | $10^{-3}$ | $10^{-3}$ |
| Batch size | 32 | 32 | - | - |
| Batch gradient | 128 | 128 | - | - |
| Dropout (classifier) | 0 | 0 | 0.5 | 0.5 |
| Epochs | 50 | 50 | 500 | 500 |

Table 3: Experimental hyperparameter values for our proposed model across all the four datasets.

**Dataset Information and Splits.** The datasets used in our experiments can be downloaded from the following sources: Food-101 from https://visiir.isir.upmc.fr, N24News from https://github.com/billywzh717/N24News, and BRCA and ROSMAP from https://github.com/txWang/MOGONET.

To ensure a fair comparison with previous works, we adopt the default split method detailed in Table 4. As the Food-101 dataset does not include a validation set, we partition 5,000 samples from the training set to create one, which is conistent with other baselines.

| Dataset | Modalities | Modality Types | Train | Validation | Test | Classes |
|---|---|---|---|---|---|---|
| Food-101 | 2 | Image, text | 60101 | 5000 | 21695 | 101 |
| N24News | 2 | Image, text | 48988 | 6123 | 6124 | 24 |
| ROSMAP | 3 | mRNA, miRNA, DNA | 245 | - | 106 | 2 |
| BRCA | 3 | mRNA, miRNA, DNA | 612 | - | 263 | 5 |

Table 4: Statistics for the four datasets: Food-101, N24News, ROSMAP, and BRCA. Note: miRNA stands for microRNA, and mRNA stands for messenger RNA.

## A.2 Comparison with Baselines

We compare our method with various multimodal classification approaches [25, 27, 32–50]. Some methods [38, 40, 41] focus on integrating global features from individual modality-specific backbones to enhance classification. Others [42–45] use sophisticated pre-trained architectures fine-tuned for specific tasks. UniS-MMC [36], the previous state-of-the-art on Food-101 and N24News, uses contrastive learning to align features across modalities with supervision from unimodal predictions. Similarly, Dynamics [35], the previous state-of-the-art on ROSMAP and BRCA, applies a dynamic multimodal classification strategy. On Food-101 and N24News, we compare against baseline unimodal networks (ViT and BERT/RoBERTa) and our UniConcat baseline, where pre-trained image and text encoders are fine-tuned independently, and the unimodal representations are concatenated for classification.

The results are reported as the average and standard deviation over three runs on Food-101/N24News, and five runs on ROSMAP/BRCA. The best score is highlighted in bold, while the second-best score is underlined. The classification accuracy on N24News and Food-101 are displayed in Table 1 and 5 respectively. In the result tables, **ALI** denotes alignment (indicating if the method employs a contrastive component), while **AGG** specifies whether aggregation is early (combining unimodal feature) or late fusion (combining unimodal decisions).

The experimental results from Table 1, 2, 5, reveal the following findings: **(i)** M3CoL consistently outperforms all SOTA methods across all text sources on N24News when using the same encoders, beats SOTA on all evaluation metrics on ROSMAP and BRCA, and also achieves competitive results on Food-101; **(ii)** contrastive-based methods with any form of alignment demonstrate superior performance compared to other multimodal methods; **(iii)** our proposed M3CoL method, which employs a contrastive-based approach with shared alignment, improves over the traditional contrastive-based models and the latest SOTA multimodal methods.

| Method | Fusion | | Backbone | | ACC $\uparrow$ |
|---|---|---|---|---|---|
| | AGG | ALI | Image | Text | |
| Image-only | - | - | ViT | - | 73.1 |
| Text-only | - | - | - | BERT | 86.8 |
| UniConcat | Early | ✗ | ViT | BERT | 93.7 |
| MCCE | Early | ✗ | DenseNet | BERT | 91.3 |
| CentralNet | Early | ✗ | LeNet5 | LeNet5 | 91.5 |
| GMU | Early | ✗ | RNN | VGG | 90.6 |
| ELS-MMC | Early | ✗ | ResNet-152 | BOW features | 90.8 |
| MMBT | Early | ✗ | ResNet-152 | BERT | 91.7 |
| HUSE | Early | ✓ | Graph-RISE | BERT | 92.3 |
| VisualBERT | ✗ | ✓ | FasterRCNN+BERT | BERT | 92.3 |
| PixelBERT | Early | ✓ | ResNet | BERT | 92.6 |
| ViLT | Early | ✓ | ViT | BERT | 92.9 |
| CMA-CLIP | Early | ✓ | ViT | BERT | 93.1 |
| ME | Early | ✗ | DenseNet | BERT | **94.7** |
| UniS-MMC | Early | ✓ | ViT | BERT | **94.7** |
| M3CoL (Ours) | Early | ✓ | ViT | BERT | $\underline{94.3}_{\pm 0.04}$ |

Table 5: Classification Accuracy (ACC) comparison on Food-101. AGG denotes early/late modality fusion, ALI indicates presence/absence of alignment.

### A.3  Analysis of Our Method

**Effect of Vanilla Mixup.** Mixup involves two main components: the random convex combination of raw inputs and the corresponding convex combination of one-hot label encodings. To assess the performance of our M3CoL method in comparison to this Mixup strategy, we conducted experiments on the Food-101 and N24News datasets (text source: abstract). We remove the contrastive loss from our framework (Eq. 11) while keeping the rest of the modules unchanged. Table 6 shows that the **Mixup** technique underperforms relative to our proposed M3CoL approach. The observed accuracy gap can be attributed to excessive noise introduced by label mixing, and the lack of a contrastive approach with an alignment component. This indicates that the vanilla Mixup strategy introduces additional noise which impairs the model's ability to learn effective representations, while our M3CoL framework benefits from the structured contrastive approach.

**Effect of Loss & Unimodality Supervision.** To assess the necessity of each component in the framework, we investigate several design choices: (i) the framework's performance without the supervision of unimodal modules during training, and (ii) the performance differences between using only MultiSClip and only M3Co loss during end-to-end training. The M3CoL (**No Unimodal Supervision**) result indicates that excluding the unimodal prediction module results in a decline in performance as shown in Table 6, highlighting its importance as it allows the model to compensate for the weaknesses of one modality with the strengths of another. Additionally, the M3Co loss (**only**

**M3Co**) outperforms the MultiSClip loss (**only MultiSClip**) by learning more robust representations through Mixup-based techniques, which prevent trivial discrimination of positive pairs. Furthermore, using an individual contrastive alignment approach (**only M3Co**) throughout the entire training process without transitioning to the MultiSClip loss results in suboptimal outcomes. This can be attributed to the risk of over-training with Mixup-based loss, which may negatively impact generalization. This demonstrates the necessity of the transition of the contrastive loss during training (**0.33 M3Co + 0.67 MultiSClip**).

| Method | ACC ↑ | | | |
|---|---|---|---|---|
| | **ROSMAP** | **BRCA** | **Food-101** | **N24News** |
| Mixup | $84.13_{\pm 0.74}$ | $84.52_{\pm 0.46}$ | $93.14_{\pm 0.02}$ | $81.57_{\pm 0.24}$ |
| M3CoL (No Unimodal Supervision) | $85.14_{\pm 0.85}$ | $86.93_{\pm 0.52}$ | $94.12_{\pm 0.02}$ | $84.26_{\pm 0.11}$ |
| M3CoL (only MultiSClip) | $86.84_{\pm 0.34}$ | $87.38_{\pm 0.41}$ | $94.23_{\pm 0.01}$ | $84.06_{\pm 0.18}$ |
| M3CoL (only M3Co) | $87.42_{\pm 0.63}$ | $87.74_{\pm 0.42}$ | $94.24_{\pm 0.12}$ | $84.57_{\pm 0.08}$ |
| M3CoL (0.33 M3Co + 0.67 MultiSClip) | $\mathbf{88.67}_{\pm 0.94}$ | $\mathbf{88.38}_{\pm 0.57}$ | $\mathbf{94.27}_{\pm 0.04}$ | $\mathbf{84.72}_{\pm 0.03}$ |

Table 6: Accuracy (ACC) on ROSMAP, BRCA, N24News, and Food-101 datasets under different settings of our method. For N24News, source: abstract and encoder: RoBERTa.

## A.4 Visualization of Attention Heatmaps

The attention heatmaps generated using the embeddings from our trained M3CoL model in Figure 3 and 4 highlight image regions most relevant to the input word. We generate text embeddings for class label words and corresponding image patch embeddings, computing attention scores as their dot product. This visualization aids in understanding the model's focus, decision-making process, and association between class labels and specific image regions. Importantly, it also indicates the correctness of the learned multimodal representations, revealing the model's ability to ground visual concepts to semantically meaningful regions.

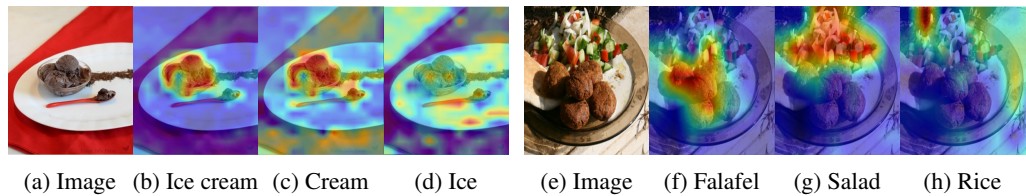

(a) Image  (b) Ice cream  (c) Cream  (d) Ice     (e) Image  (f) Falafel  (g) Salad  (h) Rice

Figure 3: Text-guided visual grounding with varying input prompts. (a, e) Original images. (b-d) Attention heatmaps for "ice cream" class. (f-h) Heatmaps for "falafel" class. Ice cream example: (b) "Ice cream": Concentrated focus on ice cream, (c) "Cream": Maintained but diffused focus, (d) "Ice": Dispersed attention. Falafel example: (f) "Falafel": Localized focus on falafel, (g) "Salad": Attention shift to salad component, (h) "Rice": Minimal attention (absent in image). Warmer colors indicate higher attention scores.

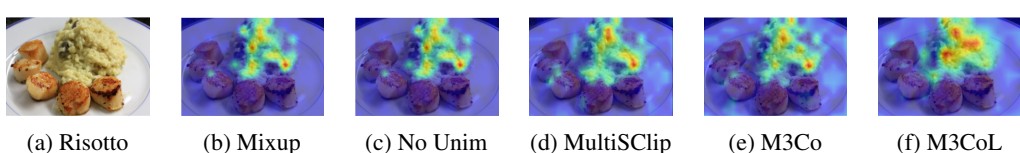

(a) Risotto     (b) Mixup     (c) No Unim     (d) MultiSClip     (e) M3Co     (f) M3CoL

Figure 4: Text-guided visual grounding with ablated model variations. (a) Original image. (b-f) Attention heatmaps generated using text embedding (class name: "Risotto") and patch embeddings for different variations of the model. Our proposed M3CoL model (f) demonstrates superior attention localization compared to ablated versions (b-e), corroborating the quantitative results presented in Table 6. Warmer colors indicate higher attention scores. (Here, No Unim: No Unimodal Supervision)

## A.5    Related Work

Approaches in multimodal learning are broadly categorized into alignment-based methods, which capture modality-invariant characteristics [51, 52], and aggregation-based techniques that combine features across modalities [53, 54]. The design of multimodal networks is typically informed by the task objective, available data, and computational constraints [55–58]. Common strategies include inputting all modalities as token embeddings, performing cross-attention between modalities, concatenating representations, and ensemble-based combination of modality-specific predictions [1].

**Contrastive Learning.** Contrastive learning has driven significant progress in unimodal and multimodal representation learning by distinguishing between similar (positive) and dissimilar (negative) pairs. In multimodal contexts, cross-modal contrastive techniques align representations from different modalities [2, 59, 60], with approaches like CrossCLR [61] and GMC [62] focusing on global and modality-specific representations. Contrastive learning approaches for paired image-text data, such as CLIP [2], ALIGN [59], and BASIC [63], have demonstrated remarkable success across diverse vision-language tasks. Subsequent works have aimed to enhance the efficacy and data efficiency of CLIP training, incorporating self-supervised techniques (SLIP [64], DeCLIP [65]) and fine-grained alignment (FILIP [66]). The CLIP framework relies on data augmentations to prevent overfitting and the learning of ineffective shortcuts [9, 10], a common practice in contrastive learning.

**Unimodal and Multimodal Data Augmentation.** Data augmentation has been integral to the success of deep learning, especially for small training sets. In computer vision, techniques have evolved from basic transformations to advanced methods like Cutout [67], Mixup [4], CutMix [5], and automated approaches [6, 68]. NLP augmentation includes paraphrasing, token replacement [69, 70], and noise injection [71]. Multimodal data augmentation, primarily focused on vision-text tasks, has seen limited exploration, with approaches including back-translation for visual question answering [72], text generation from images [73], and external knowledge querying for cross-modal retrieval [74]. MixGen [75] generates new image-text pairs through image interpolation and text concatenation. In contrast, our proposed augmentation technique focusing on the early training phase is fully automatic, applicable to arbitrary modalities, and designed to leverage inherent shared relations in multimodal data.

**Relation to Mixup.** Mixup [4], a pivotal regularization strategy, enhances model robustness and generalization by generating synthetic samples through convex combinations of existing data points. Originally introduced for computer vision, it has been adapted to NLP by applying the technique to text embeddings [15]. Our proposed augmentation differs from Mixup in several key aspects: it is designed for multi-modal data, takes inputs from different modalities, and does not rely on one-hot label encodings. By extending the Mixup paradigm to complex, multi-modal scenarios and focusing on the early training phase, our method broadens its applicability while leveraging inherent shared relations in multimodal data.

