# OpenReview forum: "M3CoL: Harnessing Shared Relations via Multimodal Mixup Contrastive Learning for Multimodal Classification"
_NeurIPS.cc/2024/Workshop/UniReps — UniReps_

### Official Review · Reviewer_b15D · 2024-09-29
**Evaluation of paper "M3CoL: Multimodal classification using shared relationships through multimodal hybrid contrastive learning"**

**Rating:** 8
**Confidence:** 4

**Review:**

The author proposes a novel multimodal classification method called M3CoL, which introduces multimodal mixed contrastive learning to capture shared relationships beyond simple pairwise associations. This article addresses the limitations of traditional contrastive learning methods by focusing on shared multimodal relationships and applies them to classification tasks across different datasets. This method outperforms several state-of-the-art methods on multiple datasets, highlighting its importance.

---

### Official Review · Reviewer_uhto · 2024-10-04
**Paper proposes a MixUp augmentation for Multi-model contrastive learning; The proposed method seems connected to introducing label noise in the first stages of the representation learning and leads to downstream performance gains.**

**Rating:** 6
**Confidence:** 4

**Review:**

This paper proposes a Mixup-based augmentation for multimodel contrastive learning which seems quite similar to having label noise. The authors show that the proposed augmentation which is applied toward the start of the training but abandoned later on is beneficial and leads to downstream performance gains on text-image and multimodel medical datasets.

The contribution is based on heuristics and could be further developed to bring more insights into the inner workings of the proposed method that led to this performance increase (i.e., link to label noise, link to temperature parameter). That being said the topic is aligned with the interests of the audience, the paper is well-written and the idea can be interesting if further developed notably to better understand the key components behind multi-model representation learning.

The format of this submission should be improved, the 4-page format does not include the experimental results which are delegated to the appendix. Meanwhile, the explanation behind the method which I believe could hold in 2 pages takes up 3 out of the 4 pages of this submission.

---

### Official Review · Reviewer_8ubM · 2024-10-06

**Rating:** 7
**Confidence:** 3

**Review:**

## Summary
The paper introduces a method M3CoL to enhance multimodal learning by capturing shared relations between different modalities using Mixup-based contrastive learning. The M3CoL aligns mixed samples from one modality with corresponding samples from other modalities, thereby capturing nuanced shared relations inherent in multimodal data. The framework integrates a fusion module with unimodal prediction modules for auxiliary supervision during training. Extensive experiments on diverse datasets demonstrate that M3CoL effectively captures shared multimodal relations, outperforming state-of-the-art methods on several benchmarks.

## Strengths
1. The introduction of Mixup-based contrastive loss to capture shared relationships between different modalities that extends beyond traditional one-to-one contrastive learning.
2. The paper provides extensive experimental results on diverse datasets, demonstrating the effectiveness of M3CoL.

## Weaknesses
1. The performance of M3CoL may be sensitive to the choice of hyperparameters, such as the temperature parameter and the Mixup coefficient. It should be discussed.
2. It would be better to discuss the performance of the method on other types of multimodal data, such as  audio-image/video.
3. It would be better to analyze and discuss the limitations of the method and the samples that failed to predict.

---

### Official Review · Reviewer_EFGh · 2024-10-07
**M3CoL: Promising Multimodal Mixup Approach with Room for Clarity and Improvement**

**Rating:** 7
**Confidence:** 3

**Review:**

## Overview:

This paper introduces M3CoL, a Multimodal Mixup Contrastive Learning approach for multimodal classification tasks. The method aims to capture shared relations across different samples in multimodal data by using a Mixup-based contrastive loss.

## Pros:

1. Clear motivation: The paper provides a compelling rationale for capturing shared relations in multimodal data.
2. Well Detailed Architecture: The authors present an architecture that is easy to understand and well motivated.
3. Comprehensive evaluation: The authors test their approach on diverse datasets across different domains.

## Cons:

1. Complexity: The proposed method appears quite complicated, which may make it challenging to implement or interpret. Specifically, the loss function although well detailed seems very large (O(n^2) terms for the batch size n).
2. Potential overfitting: The method of mixing potentially unrelated elements might lead to overfitting to noise, which they compensate for by a scheduled loss function.
3. Lack of clarity on novelty: The distinction between this work and previous methods (references 11-13 and MixCo) is not clearly articulated. Is it that no previous methods perform multimodal mixing?
4. Improve results presentation: Include a succinct results table in the main paper, possibly moving some detailed equations to the appendix to make room. Maybe move the full multiSClip loss to the appendix? or try to combine eq. 8 and 9 with $\mathcal{L}_\text{M3CO}^{(K)} $ for $ K \in {1,2}$ or something similar.

## Suggestions:

1. Clarify the mixing process: Explain how the method handles cases where mixed elements are completely distinct and how this affects the model's learning.
2. Consider unimodal inference: Explore whether the unimodal classifiers that were used during training can perform inference alone, which could be a valuable feature. It might be interesting to see if the unimodal methods trained through M3CoL perform the same as a regular unimodal mode, giving M3CoL the flexibility to perform with each modality individually as well.

## Typos and Clarifications Needed:

1. Figure 1: Are the +sim's mixed in the same way with the λ's?
2. Line 72: Define $m_i$, as its meaning is not clear from the context.

---

### Decision · Program_Chairs · 2024-10-10

**Decision:**

Accept

**Comment:**

In light of the positive reviewers' feedback and relevancy of the submission, we are pleased to accept this paper for presentation at UniReps 2024. We kindly ask the authors to incorporate the reviewers' suggestions and feedback in the final camera-ready version of the manuscript.